# Dopamine improves defective cortical and muscular connectivity during bilateral control of gait in Parkinson's disease
Paulo Cezar Rocha dos Santos [1,2,3] ✉, Benedetta Heimler [2], Or Koren[2], Tamar Flash[1] &
Meir Plotnik [2,4,5] ✉

Parkinson's Disease (PD)-typical declines in gait coordination are possibly explained by weakness in bilateral cortical and muscular connectivity. Here, we seek to determine whether this weakness and consequent decline in gait coordination is affected by dopamine levels. To this end, we compare cortico-cortical, cortico-muscular, and intermuscular connectivity and gait outcomes between body sides in people with PD under ON and OFF medication states, and in older adults. In our study, participants walked back and forth along a 12 m corridor. Gait events (heel strikes and toe-offs) and electrical cortical and muscular activities were measured and used to compute cortico-cortical, cortico-muscular, and intermuscular connectivity (i.e., coherences in the alpha, beta, and gamma bands), as well as features characterizing gait performance (e.g., the step-timing coordination, length, and speed). We observe that people with PD, mainly during the OFF medication, walk with reduced step-timing coordination. Additionally, our results suggest that dopamine intake in PD increases the overall cortico-muscular connectivity during the stance and swing phases of gait. We thus conclude that dopamine corrects defective feedback caused by impaired sensory-information processing and sensory-motor integration, thus increasing cortico-muscular coherences in the alpha bands and improving gait.

Impaired top–down control of movement in people with Parkinson's disease (PD) may underlie impairments in gait rhythmicity and in bilateral left–right synchronicity during gait. Empirical evidence has shown that, when compared to healthy older (OA) and younger adults, people with PD exhibit 2–10 times lower temporal coordination between left and right stepping, as is indicated by higher values of the phase coordination indexes (PCI)[1,2], with a distorted ratio between the durations of the stance and swing phases. In healthy younger adults, the proportion of stance and swing phase during gait varies from 60 to 62% versus 40–38% of the gait cycle, and the ratio is close to the "golden ratio" ($\phi \approx 1.618$—an irrational number—as an indicative of gait harmony)[1].

Despite substantial evidence indicating that stepping synchronicity is primarily governed at the spinal level by central pattern generators[3,4], studies have demonstrated substantial cortical involvement during bilateral gait coordination in PD[5]. Specific and asymmetric modulations between cortical areas and cortical and muscular activations in PD may underlie gait impairments[5–7]. For example, people with PD exhibit heightened overall brain activity[5,8–10], including bilateral cortical hypersynchronization and a consequent dysfunction in cortico-muscular connectivity during walking[6]. The bilateral cortical hypersynchronization may arise from the asymmetric neural degeneration observed in PD[9]. Such asymmetry, which reflects an uneven predominance of motor impairments between the more affected (MAS) and the less affected sides (LAS), may also be related to an asymmetrical (sub)cortical contribution to the muscles involved in gait in people with PD[1,11].

Asymmetrical neurodegeneration and loss of dopamine in the basal ganglia seemingly lead to a compensatory neural "re-wiring"[12], which might be minimized by levodopa intake. For instance, the over-activation of right and left brain cortices in PD (a utilization of neural circuits to compensate for the neural deficits caused by the disease[13,14]) is higher without medication

[1]Department of Computer Science and Applied Mathematics, Weizmann Institute of Science, Rehovot, Israel. [2]Center of Advanced Technologies in Rehabilitation, Sheba Medical Center, Ramat Gan, Israel. [3]IDOR/Pioneer Science Initiative, Rio de Janeiro, Brazil. [4]Sagol School of Neuroscience, Tel Aviv University, Tel Aviv, Israel. [5]Department of Physiology and Pharmacology, Faculty of Medicine, Tel Aviv University, Tel Aviv, Israel. ✉e-mail: paulo-cezar.rocha-dos-santos@weizmann.ac.il; paulocezarr@hotmail.com; meir.plotnik@sheba.health.gov.il

(OFF state)[10]. Furthermore, cumulative evidence shows that ingesting levodopa significantly optimizes cortical activity[8,10] and improves gait performance, making people with PD walk more similarly to healthy OA. This finding confirms the essential role of dopamine in the cortical control of gait. Thus, a greater understanding of the interaction between the asymmetrical degeneration of dopaminergic neurons in PD and the role of levodopa intake with respect to the organization and coupling of cortical and muscular activities may give rise to new plausible modes of treatment.

Because of PD-related impairments in the central nervous system, the organization and delivery of neural drives to muscles in people with PD are likely defective. One way to estimate the organization and coupling between activities at different cortical sites and muscle activation is to assess the cortico-cortical, cortico-muscular, and intermuscular coherences[15–17]. Coherence reflects oscillatory couplings between two sources by measuring the degree of association between two signals in a certain band of frequency domain (alpha—5–15 Hz, beta—16–35 Hz, gamma—36–55 Hz)[15–17]. Functionally, coherence reflects a connection/synchronicity between those two sources[18]. Interpretation of findings concerning coherence and gait is dependent on the frequency range[18]. In cortical areas, an over-synchronicity between brain hemispheres (based on electroencephalography [EEG] signals) in all frequency bands, as is reported in people with PD[9,10], is presumably associated with lower gait performance. Regarding cortico-muscular signals (EEG–electromyography [sEMG]), alpha and beta oscillations are widely recognized as the most pertinent frequency bands during motor tasks[19]. These oscillations are associated with two primary descending motor pathways: the corticospinal tract and the corticoreticulospinal tract. Additionally, they play a crucial role in processing sensory inputs and facilitating sensory-motor integration[20]. Intermuscular coherence (sEMG–sEMG) within a certain frequency range, in turn, may provide information on the origin of motor organization at the cortical/subcortical and spinal levels[21]. Studies in populations with spinal cord injuries have indicated a relatively preserved intermuscular coherence in lower- (e.g., alpha) but not in higher-frequency bands (beta and gamma) during gait, indicating that while oscillatory synchronicity in the alpha seems to be driven at spinal levels, beta and gamma bands may be related to a motor organization at subcortical levels[21].

In PD, combining analyses of cortico-cortical, cortico-muscular, and intermuscular coherence during gait may help elucidate the role of dopamine on the neural control of walking. The little and fragmented existing evidence has suggested impaired coherence in PD. In terms of cortico-cortical coherence, people with PD walk with cortico-cortical over-synchronicity[9,14]. This over-synchronicity is interpreted as a "compensatory" response to PD-related neurodegenerative processes that minimizes gait impairments (e.g., decreased step length, speed, and coordination)[9,10]. Additionally, due to PD-related dysfunction in sensory-information processing and sensory-motor integration, the organization and the output of descending motor drives are affected, reflecting weak cortico-muscular coherence in the alpha (mainly) and beta during gait[6]. It is, however, surprising that the asymmetrical prevalence of symptoms in PD's gait is not accompanied by an asymmetry in cortico-muscular connectivity[6]. It might be that asymmetrical neurodegenerative characteristics of PD may reflect asymmetrical changes in the organization of neural inputs at subcortical levels, revealing side differences in intermuscular coherence and bilateral coordination of gait.

The understanding of the relationship between gait measurements and cortico-cortical and intermuscular neural activation coupling can contribute to our understanding of the underlying neural mechanism(s) responsible for the lack of harmony, symmetry, and step-timing coordination. Thus, our aim is to compare (primary) cortico-muscular but also cortico-cortical and intermuscular connectivity and gait metrics (gait score, harmony, and step-time coordination) between the MAS and LAS in people with PD in ON and OFF medication states (PD-ON and PD-OFF, respectively). As a reference, we compare our PD-related findings to OA. We had three hypotheses: First, PD-related reduction in dopamine would affect descending motor tracts and be reflected in lower cortico-muscular

coherence, mainly in cortico-muscular in the alpha and beta bands[6]. Specifically, we expected that PD vs. OA would have weaker coherence, which would be even weaker in PD-OFF during strides of the MAS (considering the asymmetrical characteristic of the disease). Second, cortico-cortical coherence would be greater in PD-OFF due to compensation for PD-related impairments in gait control[10]. With medication intake, we expected a reduction in cortico-cortical coherence accompanied by improvements in gait outcomes (improved step-time coordination, increased gait speed, and greater step length). Third, we expected intermuscular coherence in the beta band among synergistic muscle pairs to be reduced in PD-OFF, mainly during steps with the MAS, indicating impaired supra-spinal control[21]. Complementarily, we correlated the medication effects on bilateral coherence with gait outcomes and PD symptoms.

## Results
### Participants
Initially, using an existing database, data sets from 28 people with PD and 10 healthy OA had the potential to be included. ON and/or OFF measurements could not be taken for 9 people with PD out of the 28 potential people with PD who were excluded from the study. An additional 5 people with PD and 1 OA (out of the 10 potential OA individuals) were excluded because of technical issues (data to detect gait events—heel strikes and toe-offs—were missing). Therefore, the final sample size was comprised of 14 people with PD and 9 OA.

The participants' characteristics and functional mobility (i.e., Timed Up and Go [TUG]) are shown in Supplementary Table 1 in the Supplementary Results (Supplementary Information). People with PD and OA were similar in all demographic characteristics except for age ($T_{21}$ = 2.15; $p$ = 0.04; $d$ = 0.99) and TUG ($T_{21}$ = 2.97; $p$ < 0.01). Medication significantly improved functional mobility since TUG time was reduced by ~6 s in the PD-ON vs. PD-OFF conditions ($d$ = −1.08). Between groups, PD-OFF vs. OA performed TUG ~9 s more slowly ($d$ = −1.3), while no difference in TUG between PD-ON and OA was observed. Specifically considering clinical measures, the medication decreased motor symptoms (decreases by ~9 points in the score for the motor part (III) of the Movement Disorder Society-sponsored revision of the Unified Parkinson's Disease Rating Scale [MDS-UPDRS], $T_{21}$ = 4.03; $p$ < 0.01; $d$ = 0.55).

### Gait score and PCI
We compared the gait score and PCI between Medication (PD-ON vs. PD-OFF), Group (PD-ON and PD-OFF vs. OA) and Sides (MAS vs. LAS). Figure 1a–c depicts the effects of Medication and Group on stride length, gait speed, and PCI, respectively. The main effect of Medication (analysis of variance (ANOVA) outcomes—Supplementary Table 2) indicated that PD-ON vs. PD-OFF increased the stride length by 15% and the gait speed by 17% and reduced the PCI by 1.4% ($d$ range =0.65–0.96, respectively). For the Group main effect, compared to OA, PD-OFF walked with ~20% shorter stride length and lower gait speed, and with ~3.2% higher PCI ($d$ range = 1.37–1.78, respectively). Also, compared with OA, PD-ON walked with 1.73% higher PCI ($d$ = 0.92). A full description of main and interaction effects is presented in Supplementary Table 2. There were no significative differences in the temporal gait score and $\phi$ (see Supplementary Results and Supplementary Figs. 3 and 4).

### Brain–muscular wave coherence
This subsection describes the results regarding the main effects/interactions of Medication, Side, and Group on coherence outcomes in alpha, beta, and gamma bands during the swing and stance phases. Full distributions of cortical-cortical, cortico-muscular, and intermuscular coherences during stance and swing phases across frequency ranges (0–55 Hz) are depicted in Supplementary Results and Supplementary Figs. 6–10.

Regarding cortico-cortical (C3-C4) coherence for the swing phase, only Side*Medication interaction was observed (Supplementary Table 2). Post hoc analysis indicated that while PD-ON had higher cortico-cortical coherence in the alpha band during the swing phases of the steps

**Fig. 1 | Gait score and PCI.** Means (bars) with standard deviations (error bars) and individual values (circles) for **a** stride length and **b** gait speed parameters normalized by the highest value for all participants, and for **c** PCI. For (**a** and **b**), blue and red bars and dots represent the more affected side (MAS) and less affected side (LAS), respectively, and for the OA group, the light blue and red colors represent the left and right sides, respectively. Pairs of slanted dashed lines indicate dichotomous comparisons. For (**c**), black, white, and gray bars represent the PD-OFF, PD-ON, and OA groups, respectively. Horizontal brackets indicate the differences.

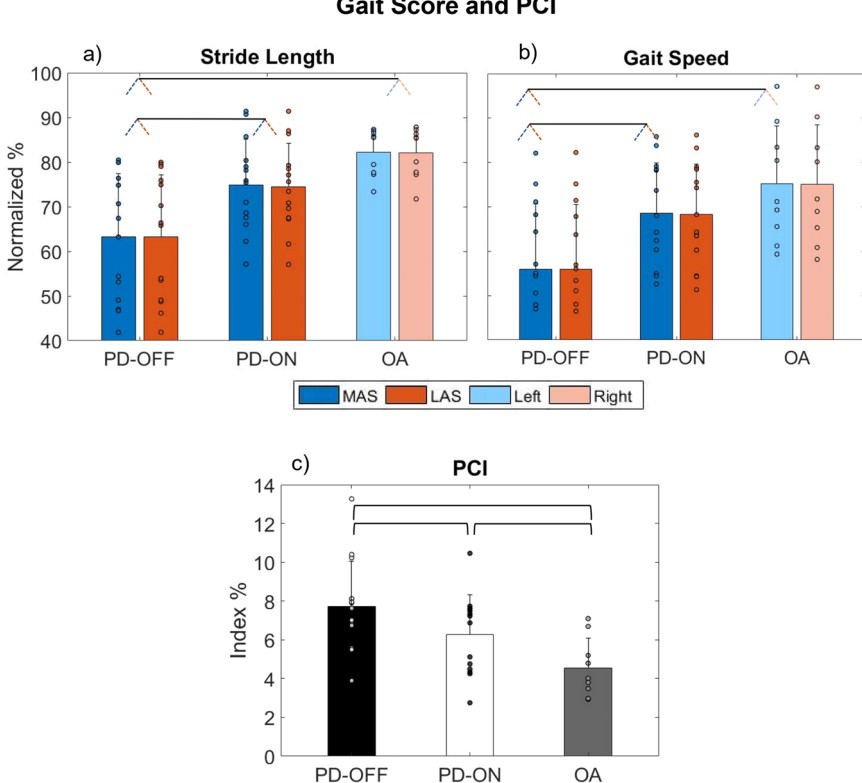

of MAS vs. LAS, PD-OFF did not have any differences between MAS and LAS ($p = 0.012$, $d = 0.30$, Fig. 2a). For cortico-cortical coherence for the stance phase, ANOVA revealed Side main effect in which PD-OFF and OA had a higher C3-C4 coherence in the gamma band in the LAS/Right side compared to the MAS/Left side ($d = 0.6$, Fig. 2b and Supplementary Table 2).

Concerning cortico-muscular coherence for the stance phase, ANOVA analyses revealed that the most prominent effect in cortico-muscular coherence is the Medication effect (i.e., among PD), particularly in the alpha band. For example, during ON vs. OFF, people with PD walked with higher cortico-muscular coherences for cortico-vastus lateralis (vastus), cortico-tibialis anterior (tibialis), and cortico-gastrocnemius lateralis (gastrocnemius) in the alpha band for the stance ($d$ range = 0.54–1.15, Fig. 3a, c, d, respectively). Additional significant effects and interactions are detailed in Fig. 3 and Supplementary Table 2.

For cortico-muscular coherence for the swing phase, the most relevant results were also observed for the Medication. Post hoc analysis indicated that PD-OFF vs. PD-ON walked with lower cortico-vastus coherence in the alpha band, cortico-biceps femoris (biceps) and cortico-tibialis coherences in the beta band, and cortico-gastrocnemius coherences in the alpha, beta, and gamma bands ($d$ range=0.43 to 1.19, Fig. 4a–d). We also observed Group differences indicating that, compared to OA, PD-ON walked with higher cortico-vastus, cortico-biceps, and cortico-gastrocnemius coherences in the alpha band ($d = 1.61$ and $2.85$, Fig. 4a, b). Additional effects are depicted in Fig. 4 and Supplementary Table 2.

Less robust effects were observed for intermuscular coherence. Herein, we describe muscles that were significantly affected (Fig. 5 and Supplementary Table 2). For the stance phase, we observed that for vastus-tibialis coherence in the alpha band, MAS of PD-OFF was lower than left (i.e., "weaker" side) steps of OA (Side*Medication interaction—Supplementary Table 2, $p = 0.031$ for the post hoc comparison, $d = 0.34$, Fig. 5a) and that PD-ON walked with higher vastus-tibialis coherence in the alpha band than did PD-OFF ($d = 0.17$, Fig. 5a). For the swing phase, ANOVA revealed a main effect of Medication, indicating that PD-ON walked with higher

biceps-gastrocnemius coherence in beta and gamma bands than did PD-OFF ($d = 0.41$ and $0.49$. respectively, Fig. 5d and Supplementary Table 2). Additional Groups, Sides, and Side*Group differences are depicted in Supplementary Table 2 and Fig. 5a–d. Supplementary Results and Supplementary Fig. 5a–d show the intermuscular coherence outcomes that ANOVA did not reveal statistical differences.

## Correlation between coherence measures and clinical–functional measures

Figure 6 summarizes the results of the correlation analyses for those outcomes that indicated Medication effects. Of note, calculating the Spearman's correlation produced a positive, significant, strong correlation, indicating that decreased PCI due to medication (indicative of better coordination) was associated with medication-induced lower biceps-gastrocnemius coherence in the beta band in the swing phase (rho = 0.61, $p = 0.018$). Moderate but not significant correlations indicated that increases in stride length and speed due to medication were negatively associated with cortico-cortical coherence in the gamma (rho = −0.47 and −0.48, respectively) and positively associated with cortico-tibialis coherence in the alpha band in the swing phase (rho = 0.50 and 0.41, respectively). In addition, an increase in stride length was moderately, but not significantly, associated with cortico-gastrocnemius coherence in the alpha band in the swing phase (rho = 0.47) and cortico-gastrocnemius coherence in the beta band in the stance phase (rho = 0.40), and negatively associated with biceps-gastrocnemius gamma in the swing phase (rho = −0.40).

Regarding clinical characteristics, Spearman's correlation revealed that Δ MDS-UPDRS-III was moderately correlated with Δ cortico-tibialis coherence in the beta band in the stance phase (rho = 0.58, $p = 0.028$). Levodopa equivalent daily dose (LEDD) was also positively associated with Δ cortico-gastrocnemius coherence in the alpha in the swing phase (rho = 0.54, $p = 0.048$) and cortico-biceps coherence in the beta in the stance phase (rho = 0.55, $p = 0.043$), and was negatively associated with Δ biceps-gastrocnemius coherence in the beta band in the swing phase (rho = 0.55, $p = 0.042$). The increased time during TUG was positively associated with Δ

## Cortico – Cortical Coherence

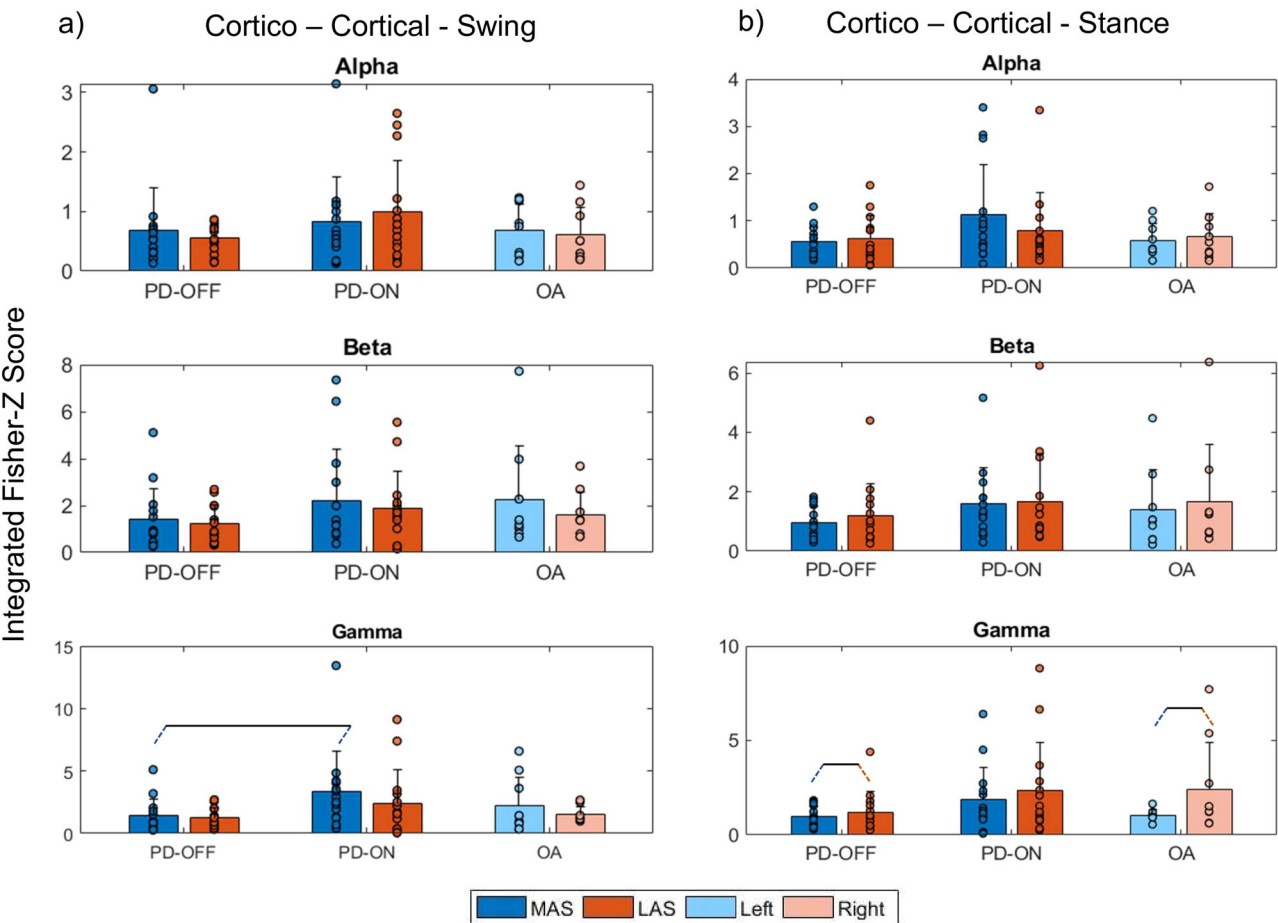

**Fig. 2 | Cortico-cortical coherence.** Means (bars) with standard deviations (error bars) and individual values (dots) for cumulative cortico-cortical (C3–C4) coherence for alpha, beta, and gamma frequency bands during **a** swing and **b** stance phases for the PD-OFF, PD-ON, and OA groups. In the figure, dark blue and red bars represent the more affected side (MAS) and less affected side (LAS), and light blue and red bars correspond to the left and right sides, respectively. Pairs of slanted dashed lines indicate dichotomous comparisons.

biceps-gastrocnemius coherence in the beta band in the swing phase (rho = 0.56, $p = 0.037$).

In order to infer the relevance of coherence in terms of functional/clinical meaning in PD, we correlate absolute coherence values during both stance and swing phases with absolute score/values of symptoms of PD, i.e., MDS-UPDRS-III, TUG, and New Freezing of Gait Questionnaire (N-FOGQ) only during OFF medication state (Fig. 7). For the stance, Spearman's correlation indicated negative moderate, but not significant ($p > 0.05$), associations between MDS-UPDRS-III and cortico-cortical coherence in the alpha (rho = −0.49), cortico-biceps coherence in the gamma (rho = -0.49), and tibialis-gastrocnemius coherence in the gamma (rho = −0.46) bands. Additionally, higher N-FOGQ was positively associated with cortico-gastrocnemius (rho = 0.58, $p = 0.031$) and with vastus-tibialis (rho = 0.42, but not significant $p > 0.05$) coherence in the gamma band. Lower TUG was associated (moderately to strongly) with greater cortico-cortical coherence in the gamma, with all cortico-muscular coherences (rho ranges from −0.49 to −0.73), with vastus-biceps coherence in the alpha beta and gamma (rho range from −0.41 to −0.62), with tibialis-gastrocnemius and biceps-gastrocnemius coherence in the alpha and gamma (rho range from −0.45 to −0.57), and with vastus-tibialis coherence in the beta (rho = -0.45) bands (Fig. 7). For the swing phase, lower scores in MDS-UPDRS-III (lower symptom manifestations) were negatively associated with cortico-vastus coherence in the gamma (rho = −0.54, $p = 0.045$) and with cortico-gastrocnemius coherence in the gamma (rho = −0.47,

$p > 0.05$) band. Additionally, cortico-muscular coherences for all pairs and bands, vastus-biceps coherence in the beta and gamma, vastus-tibialis coherence in the beta and gamma bands, and biceps-gastrocnemius coherence in the gamma band were associated with a lower time to perform TUG (rho range from −0.40 to −0.79) (Fig. 7).

## Discussion

We first compared the coherence and gait outcomes between Sides, Medication states, and Groups. Figure 8 shows a summary of the results. Our main observation was that the dopamine intake strengthened cortico-muscular coherence in the alpha bands accompanied by a decrease in PD-typical-related impairments in gait outcomes, i.e., increases in stride length and speed and improvements in step-timing coordination, making PD-ON walk more similarly to OA. Unexpectedly, in general, there were almost no Side effects on gait outcomes and cortico-muscular and intermuscular coherences, and in particular, there were few substantial results involving cortico-cortical coherences. In terms of correlations between medication-induced changes in coherences and gait scores, we observed that changes in specific outcomes of coherence were positively associated with changes in stride length and gait speed and negatively associated with PCI (e.g., decreased biceps-gastrocnemius coherence in the beta in the swing phase, with lower PCI, better coordination, and shorter strides; and increased cortico-tibialis and cortico-gastrocnemius coherence in the alpha in the swing phase with longer strides and faster walking). Overall, our

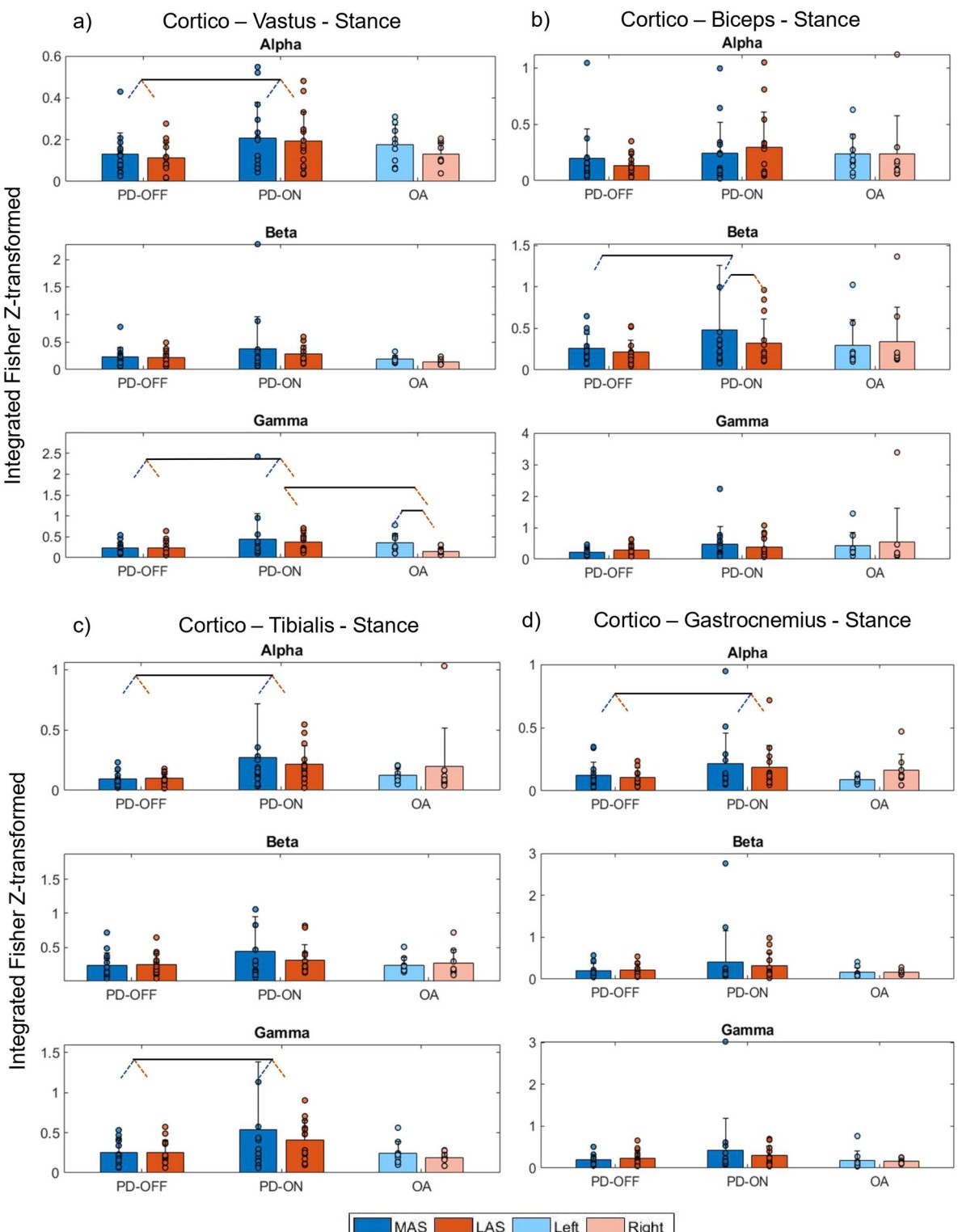

**Fig. 3 | Cortico-muscular coherence—stance.** Means (bars) with standard deviations (error bars) and individual values (dots) for cumulative cortico-muscular coherence for alpha, beta, and gamma frequency bands for the PD-OFF, PD-ON, and OA between **a** cortico-vastus lateralis; **b** cortico-biceps femoris; **c** cortico-tibialis anterior; and **d** cortico-gastrocnemius lateralis during stance phase. In the figure, dark blue and red represent the more affected side (MAS) and less affected side (LAS), and light blue and red correspond to the left and right sides, respectively. Pairs of slanted dashed lines indicate dichotomous comparisons. Note that for improving visualization, values exceeding the scale (*y*-axis) are not represented in this figure, specifically **c** beta: PD-ON MAS (value = 3) and **d** alpha and gamma: PD-ON MAS (value = 1.5 and 3.3, respectively).

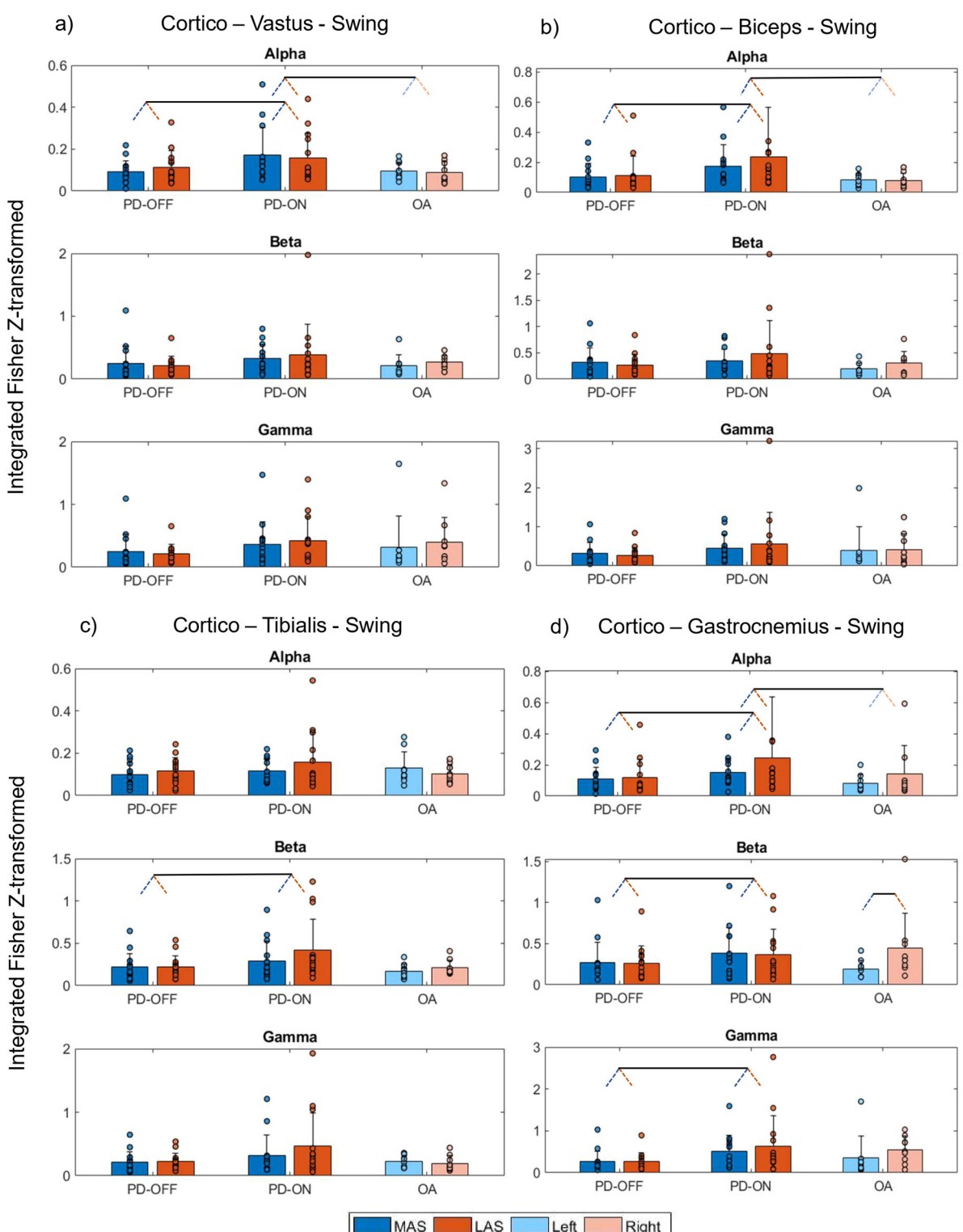

**Fig. 4 | Cortico-muscular coherence—swing.** Means (bars) with standard deviations (error bars) and individual values (dots) for cumulative cortico-muscular coherence for alpha, beta, and gamma frequency bands for the PD-OFF, PD-ON, and OA between the following pairs during swing phase: **a** cortico-vastus lateralis; **b** cortico-biceps femoris; **c** cortico-tibialis anterior; and **d** cortico-gastrocnemius lateralis. In the figure, dark blue and red bars represent the more affected side (MAS) and less affected side (LAS), and light blue and red bars correspond to the left and right sides, respectively. Pairs of slanted dashed lines indicate dichotomous comparisons. Note that for improving visualization, values exceeding the scale (y-axis) are not represented in this figure, specifically for **c** beta: PD-ON LAS (value = 1.3) and **d** alpha: PD-ON LAS (value = 1.5).

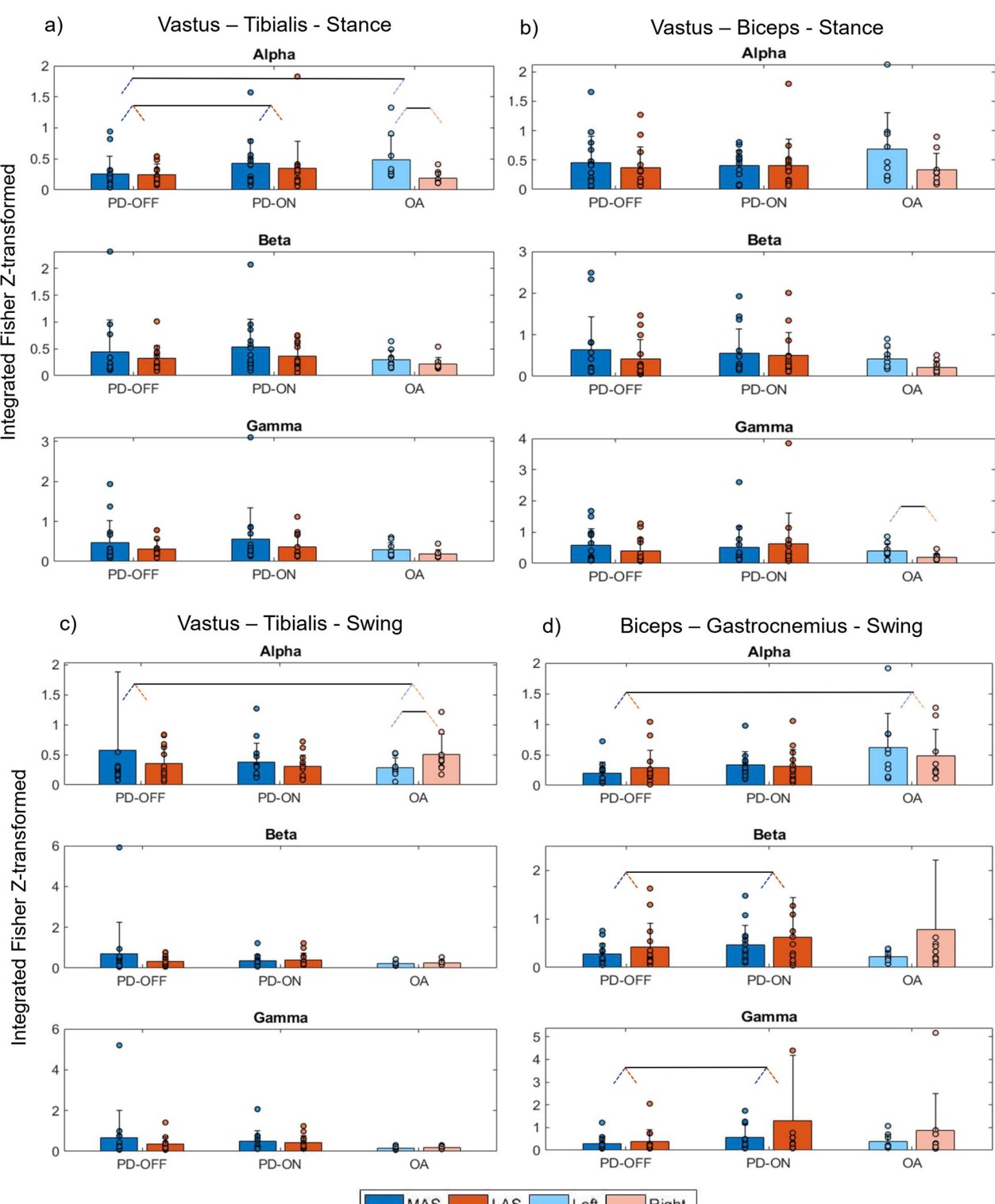

**Fig. 5 | Intermuscular coherence.** Means (bars) with standard deviations (error bars) and individual values (dots) for cumulative intermuscular coherence for alpha, beta, and gamma frequency bands for the PD-OFF, PD-ON, and OA between the following pairs: **a** vastus lateralis-tibialis anterior during stance; **b** vastus-biceps femoris during stance; **c** vastus-tibialis during swing; and **d** biceps-gastrocnemius lateralis during swing. In the figure, dark blue and red bars represent the more affected side (MAS) and less affected side (LAS), and light blue and red bars correspond to the left and right sides, respectively. Pairs of slanted dashed lines indicate dichotomous comparisons. Note that for improving visualization, values exceeding the scale (y-axis) are not represented in this figure, specifically for (**c**) beta—PD-ON LAS (value = 3.2), OA right (value = 4.6)—and (**d**) alpha—PD-OFF MAS (value = 5).

interpretation of the results was that dopaminergic medication improves gait performance, presumably due to medication-induced strengthening of the cortico-muscular connectivity.

Our results indicating no Group*Side interactions in stride outcomes (gait score) in PD is unexpected, in light of previous evidence[22] (e.g., ~6 times more gait asymmetry in people with PD than in OA[1]). Potential methodological differences may explain this discrepancy. While Plotnik et al.[22] and Iosa et al.[1] used an index to compute asymmetry, we were particularly interested in identifying whether Side differences would interact with medication to verify the role of dopamine in lateral gait control. Although

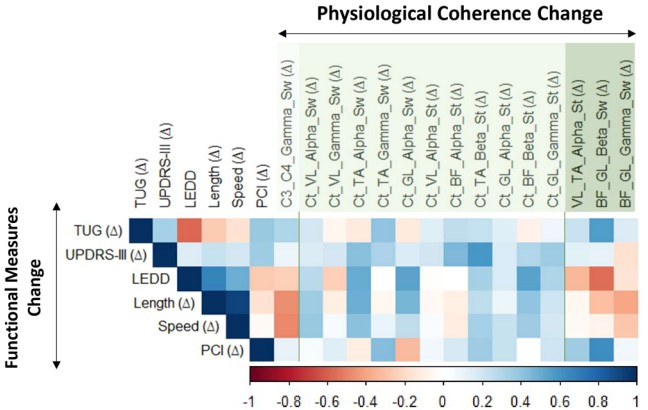

**Fig. 6 | Medication effects correlogram.** Correlations between Δ of medication state (ON–OFF) in functional measures and physiological coherence measures only for outcomes for which ANOVA indicated significant main effects of Medication. BF biceps femoris, C3-C4 cortico-cortical, Ct cortico, GL gastrocnemius lateralis, LEDD Levodopa equivalent daily dose, PCI Phase Coordination Index, St Stance, Sw Swing, TA tibialis anterior, TUG Timed Up and Go, UPDRS-III Unified Parkinson's Disease Rating Scale - motor part, VL vastus lateralis.

unexpected, the absence of Side effects on stride outcomes in PD was also accompanied by the finding indicating no differences in gait harmony ($\phi$—Supplementary Fig. 4). Therefore, in contrast to the literature[1,22], it is likely that our PD participants had an unusually symmetric and harmonic gait. Such asymmetry might be accentuated in more challenging gait conditions, e.g., dual-task, obstacle avoidance, or turning tasks[23].

Although we observed an unexpected symmetrical gait performance (e.g., swing and stance times, $\phi$), bilateral timing coordination and stride outcomes were affected by PD, mainly during the OFF state. As expected, compared to PD-OFF, our results indicated a hierarchical longer stride length, faster gait, and greater step-timing coordination in PD-ON (+15%, +17%, and −1.4%, respectively) and in OA (+20%, +20%, and −3.2%, respectively) (Figs. 1 and 2), corroborating with previous findings which reported group[1,2,22] and medication effects[11]. These results indicate that the loss of dopamine in the basal ganglia (characteristic of PD mainly during the OFF state) is reflected in hypokinesia (reduced amplitude of movement) and bradykinesia (reduced velocity), and impairs the bilateral timing coordination.

Our main observation was the medication-enhanced cortico-muscular alpha coherences in PD, occurring in both swing and stance phases. Specifically, for swing, medication intake was also reflected in increased cortico-gastrocnemius in people with PD. Surprisingly, the differences in coherences between MAS and LAS for both the swing and stance phases were small. While, on the one hand, this general absence of side effects on coherence is unexpected, on the other hand, these are in line with the observed absence of differences in gait scores for each side.

Our primary hypothesis stated that PD-related reduction in dopamine would impair top-down control of movement and would be reflected in alterations in cortico-muscular coherence. Our results confirmed this hypothesis to a certain extent. Our results indicated a defective cortical-muscular coherence in PD-OFF that improved under medication in both the swing and stance phases (Figs. 3 and 4). These results suggest that (1) PD affects cortico-muscular communication during walking, and (2) the dopaminergic medication improves the strength of the oscillatory coupling between the motor cortex and muscles. Substantial evidence suggests that

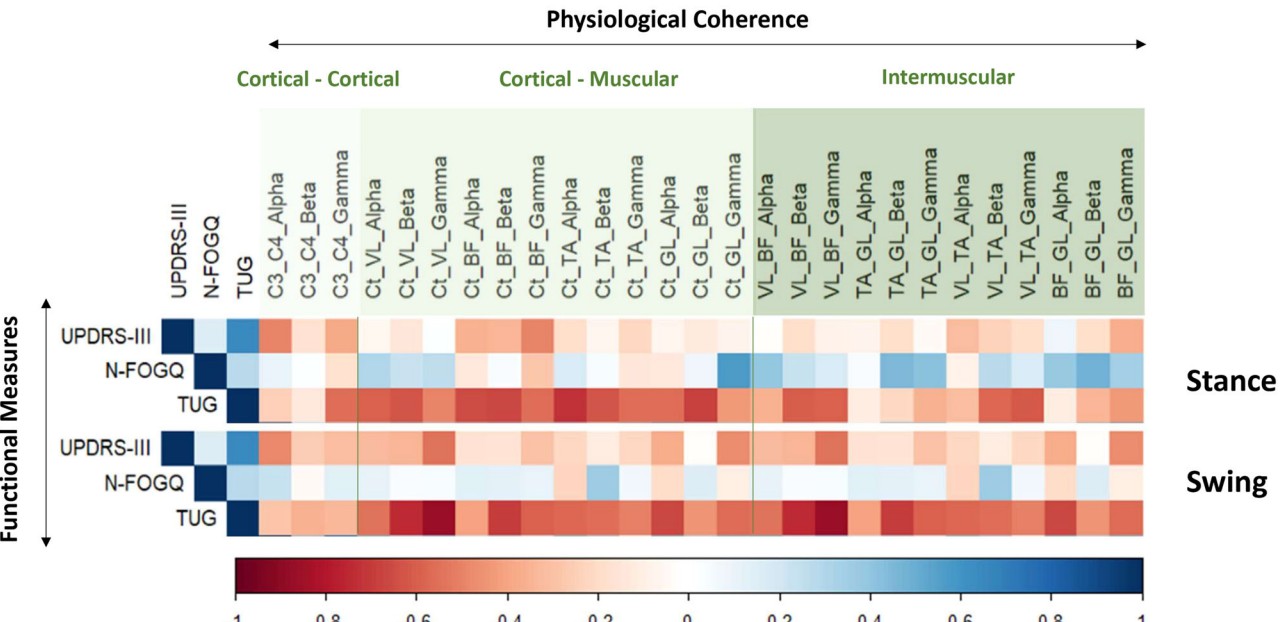

**Fig. 7 | Physiological coherence correlogram.** Correlations between physiological coherence with functional measures (clinical characteristics, functional mobility, and FoG questionnaire—N-FOGQ) in people with PD during OFF medication state. BF biceps femoris, C3-C4 cortico-cortical, Ct cortico, GL gastrocnemius lateralis, N-FOGQ New Freezing of Gait Questionnaire, TA tibialis anterior, TUG Timed Up and Go, UPDRS-III Unified Parkinson's Disease Rating Scale - motor part, VL vastus lateralis.

## Summary of Comparative Results

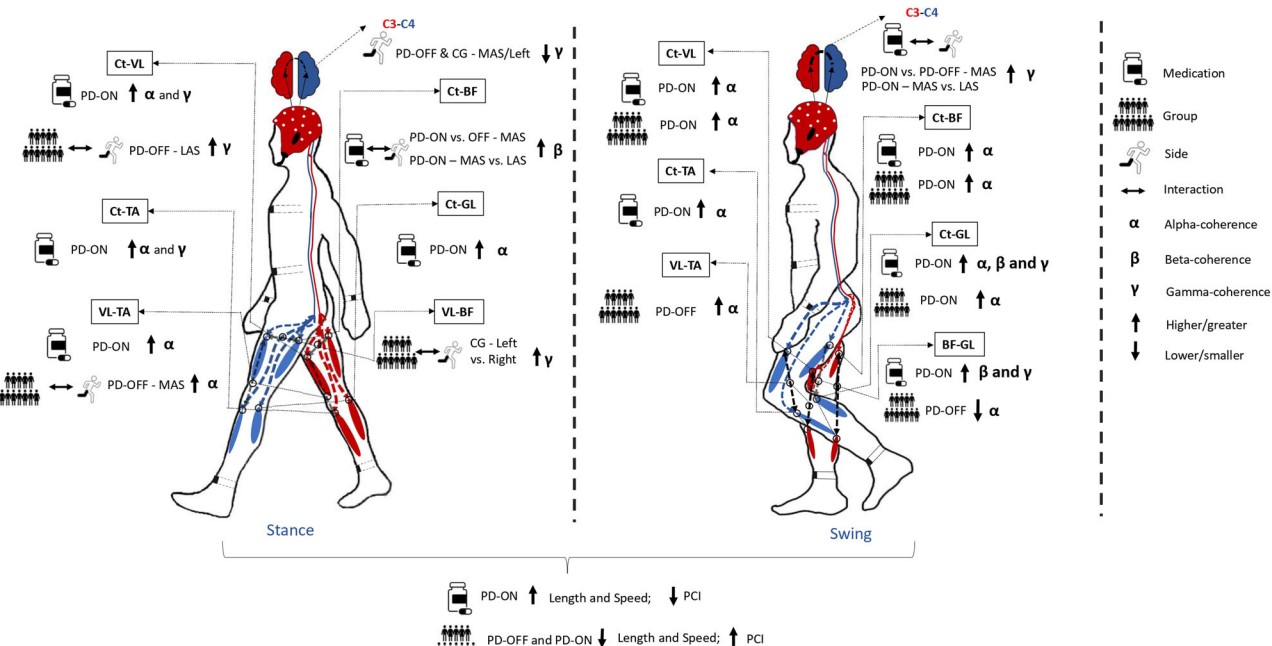

**Fig. 8 | Summary of comparative results.** Note that for Medication comparison, differences presented for PD-ON were relative to PD-OFF; for Group comparison, differences indicated for PD-ON or PD-OFF were relative to the OA group; and, for Side, differences indicated for MAS/left were relative to LAS/right. BF biceps femoris, C3-C4 cortico-cortical, Ct cortico, GL gastrocnemius lateralis, TA tibialis anterior, VL vastus lateralis. Figure adapted from Spedden et al.[17].

PD impairs the organization of motor drives to muscles[6,24]. The most documented change is the accentuated antagonistic muscle coactivation in PD, presumably due to the impaired neuromuscular control caused by the deficits in the basal ganglia[24], its connections (e.g., cortico-basal ganglia connectivity), and related areas (e.g., the pedunculus pontine nucleus)[25]. Additionally, PD-related atrophy in brain areas, such as the supplementary motor area (SMA) and M1 gray matter, reflects deficits in functionality and, thus, may explain the reduction in cortico-muscular coherence during gait. This interpretation is strengthened by the correlation results (Fig. 7), indicating that clinical impairments (higher score in MDS-UPDRS-III) and poor functional mobility in PD-OFF were associated with lower cortico-muscular coherence.

The strength of cortico-muscular coherence for the alpha band increased with dopamine medication, in some cases above the average observed in OA (e.g., Fig. 4b, d). Although we hypothesized an increase in cortico-muscular coherence after the dopamine ingestion, we did not expect this increase to be higher than in the OA. Potential explanations for such results are, to some extent, elusive since the observed increase in cortico-muscular coherence, being higher in PD than in OA, was not accompanied by the same range of increases in stride lengths and speed (Fig. 1). Thus, we can explain this finding by noting that although dopamine significantly enhances cortico-muscular connectivity in PD, the degree to which this enhancement translates into improved gait performance is constrained, resulting in performance levels that remain distinct from those observed in OA. In other words, in order to improve functionality in PD, coherence must increase substantially, but the gait will still be partially impaired.

It is important to highlight that the medication-induced increases in cortico-muscular coherence were mainly evident for activities within the alpha and only in specific cases for the beta and gamma bands. Cortico-muscular coherence in the alpha band is thought to reflect an effective functional role of sensorimotor integration (by involving both afferent and efferent systems) during the performance of a motor task[26,27] and is strongly related to the effects of sensory feedback[6,28]. Thus, the consistent lower coherence between motor cortices and the muscles in the alpha band (- Figs. 3 and 4) may be interpreted as resulting from abnormal sensory-motor integration and sensory feedback information during PD's gait in the OFF state[6]. The overall increase in cortico-muscular coherence in the alpha band after medication may indicate that dopamine improves the processing of sensory feedback information due to its positive effects on the basal ganglia and its connections with related brain areas, thus facilitating sensory-motor integration. This observation, combined with our correlation results (e.g., increased cortico-tibialis and cortico-gastrocnemius alpha correlated with increased length and speed, Fig. 6), also strengthens the understanding of the neuro-mechanisms related to the medication-induced improvement in motor performance.

We observed that dopaminergic effects on increasing cortico-muscular coherence in the beta band were evident only for the swing phase and for the coherence between cortico-ankle muscles (cortico-tibialis and cortico-gastrocnemius). The strengthening of cortico-muscular coherence in the beta band implies a higher contribution of oscillatory corticospinal activities, presumably via the corticospinal track, to the ankle muscles' activities[17,29]. Since the corticospinal track is the path related to the communication of descending motor commands to the activation of distal muscles during movements, medication-related results involving ankle muscles are reasonable to some extent. It is also plausible that, by affecting functional and structural brain areas, PD may impair cortical oscillatory contributions of (sub)cortical commands to the ankle muscles, which are the main contributors to the mechanical work performed during gait[30] and to gait speed[31]. Also, since these muscles are responsible for stabilizing the feet and keeping the toe's clearance, this impairment would explain the coherence in the beta band during the swing but not the stance phases.

To support the understanding of potential PD interference with top-down control of gait, cortico-cortical coherence would provide us with additional information on interhemispheric connectivity in motor-cortical regions. We had expected a decrease in interhemispheric connectivity. Contrary to our hypothesis, we observed an increase in cortico-cortical coherence in the gamma band (interhemispheric connectivity) during the swing phase for the MAS in PD-ON (Fig. 2a). This result is also contrary to previous evidence that indicated a decreased interhemispheric synchronization in the overall brain activity band (theta, alpha, beta) during PD

## Experimental Design – Gait, EEG & sEMG

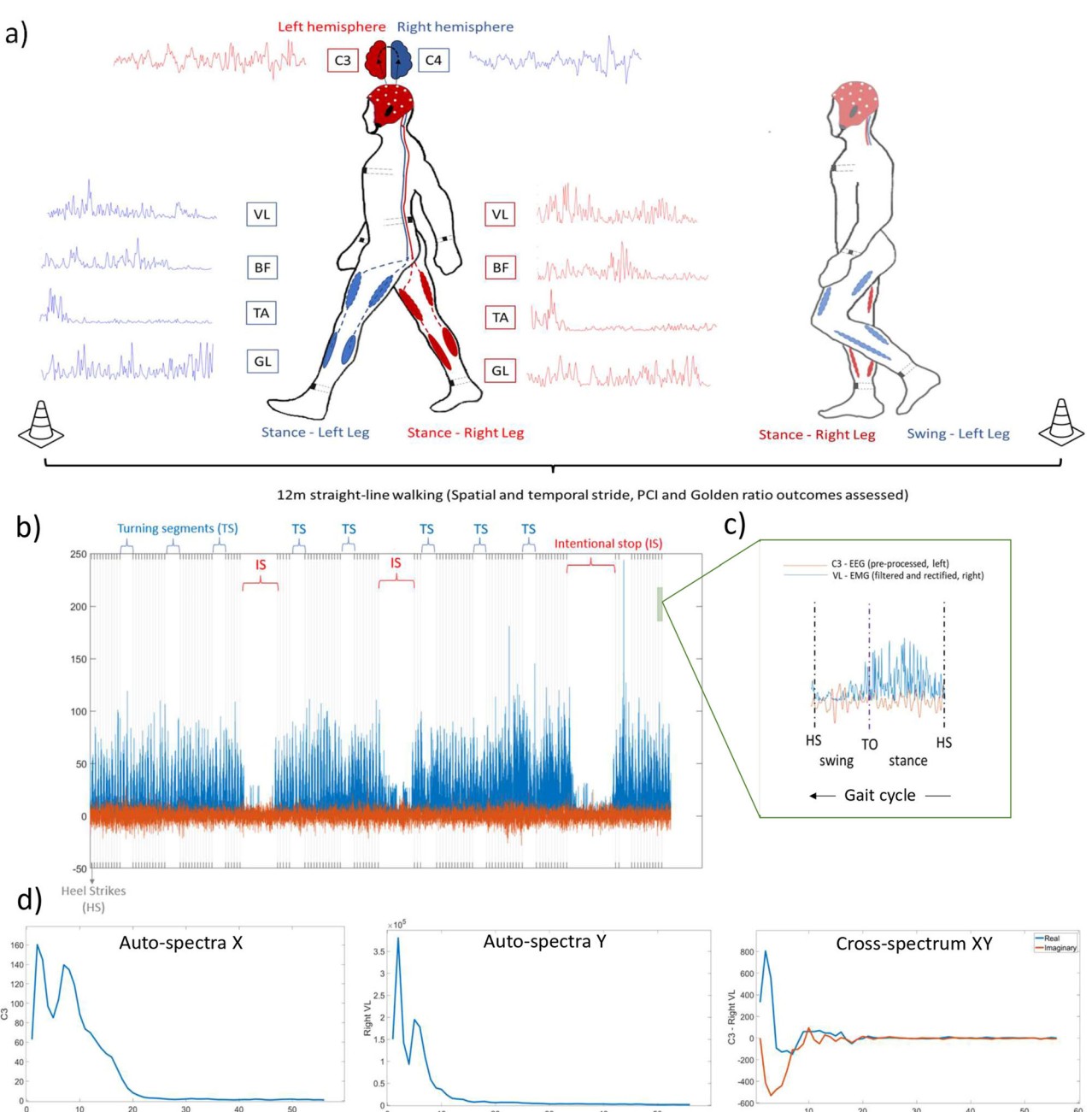

**Fig. 9 | Experimental design—gait EEG and sEMG. a** Experimental design. EEG and sEMG activities, as well as spatial and temporal gait outcomes, were collected during 12 m straight-line walking. Cortico-cortical, cortico-muscular and intermuscular coherences were computed for the stance and swing phases. Specifically, considering cortico-muscular coupling, cross coherences were computed between the right brain hemispheres (C4) and left leg muscles (vastus lateralis (VL), biceps femoris (BF), tibialis anterior (TA), and gastrocnemius lateralis (GL)) [indicated by blue] and between the left-brain hemisphere (C3) and the right leg muscles (VL, BF, TA, and GL) [indicated by red]. **b** A participant data acquisition of C3 (red line) and VL of the right side (blue line) of ~120 strides based on right heel strikes (HS—dashed vertical lines) during continuous walking. Small and bigger gaps indicate the turning segments of the walking and intentional stops (to control for potential freezing of gait) that were excluded from the analysis. **c** A zoom-in of a right gait cycle for better visualization in which it is possible to see the toe-off (TO) used to determine gait phases (swing and stance). **d** Auto-spectra for C3 (X) and VL of the right leg (Y) and cross-spectrum XY for further computation of coherence. Figure 9a adapted from Spedden et al.[17].

walking under medication effects[10]. Such differences between our results and previous evidence might be attributed to methodological aspects. Unlike our study, Koren et al.'s study[10] did not compute connectivity for specific gait phases and strides for MAS and LAS. While such methods allowed us to have a higher temporal resolution and differentiate between sides due to PD, Koren et al.'s study[10] had a more representative spatial resolution analysis of brain activities, as it combined multiple electrodes to better represent different brain areas.

Collectively, reports in the literature have argued that the hypersynchronization of the two hemispheres in the OFF state arises in order to compensate for the reduced subcortical input to the cortex due to basal ganglia dopaminergic loss[10]. Such observation seems to be in line with the

https://doi.org/10.1038/s42003-024-06195-5 **Article**

HAROLD (Hemispheric Asymmetry Reduction in Older Adults) model[32] and with the interpretation that dopamine would optimize brain function and improve asymmetrical brain control during motor tasks[33]. We opposingly observed higher C3–C4 (i.e., cortico-cortical) coherence in the gamma band (potentially increased symmetry) only for steps for the MAS in the ON state. A comprehensive explanation for this phenomenon remains elusive, even more so because our correlation results indicated that higher cortico-cortical coherence in the gamma band was associated with shorter and slower strides (Fig. 6), similar to previous reports (lower connectivity = better motor performance in PD)[10,33]. If MAS alone is considered, this association was even higher—increased cortico-cortical coherence in the gamma band correlated with shorter and slower strides (rho = 0.58 and 0.54, $p$s > 0.05), supporting the suggestion that lower hemispheric connectivity represents better gait performance.

Our results indicated that dopaminergic medication increased vastus-tibialis alpha and biceps-gastrocnemius coherences in the beta and gamma bands (Fig. 5a, c). Evidence suggests that intermuscular coherence between synergistic muscles indicates that both muscles share a common presynaptic input[15–17,34]. For vastus-tibialis coherence in the alpha, that input seems to have a lesser cortical and more spinal origin, while biceps-gastrocnemius in beta and gamma (higher-frequency) activities, the intermuscular coherence is indicated to have higher cortical involvement via pyramidal tract contribution[35]. In synergistic muscles, increasing the strength of shared common input may be interpreted as an attempt of the central nervous system to reduce the dimensionality and complexity of muscular control during the performance of a motor task[36]. However, biceps femoris and gastrocnemius do not necessarily have the same functional roles during gait[37]. Whereas biceps femoris has an important role in flexing the knee during the swing phase, gastrocnemius is probably the prime mover responsible for the push-off, as it is active during different times of the gait cycle[37]. Therefore, increased biceps-gastrocnemius coherences in the beta and gamma bands in PD may impair rather than help gait by representing an unnecessary resource for controlling oscillatory muscle coupling, reflecting rigidity. This interpretation is particularly reasonable considering that a medication-induced increase in biceps-gastrocnemius coherence in the beta band correlated with worse step-timing coordination (higher PCI values) and lower functional mobility (increase in TUG) (Fig. 6).

Although we intended to include 38 participants (28 with PD and 10 OA), only 23 (14 with PD and 9 OA) met the inclusion criteria. This relatively small sample size may have minimized potential groups/medication differences and may have affected correlation analyses. While our correlation analyses (Figs. 6 and 7) suggest moderate associations with potential clinical relevance, caution in interpretation is warranted due to the small sample size and the absence of correction for multiple correlations (as correlations here have exploratory characteristics). Future studies exploring the clinical implications of coherence analysis should prioritize larger sample sizes and rigorous statistical criteria. Furthermore, regarding the subjects' population, OA were older than PD. As coherence indicates sensitivity to age[16,17], the age difference between the two groups may have interfered with our comparisons at the group level. However, even with the age difference, performance in terms of gait scores and PCI were considerably "better" in OA vs. PD. Methodological concerns should also be considered. Firstly, EEG is highly sensitive to movement artifacts that can affect the analysis. We tried to minimize those artifacts using robust methods[38–42] and to exclude the heel strike and toe-off periods from the analyses[17]. As we computed coherence considering only two EEG electrodes, this computation might have increased the risk of cross-talk between C3 and C4 channels. We, therefore, tried to reduce this risk by checking the cumulant density individually. In addition, by performing EEG analysis in a re-scaled time window (200 data points), we avoided the heterogeneity of swing and stance segments within and between groups affecting the data, but this procedure may have obscured potential differences and comparisons with the literature, as well as temporal differences in coherence within each segment.

It is also possible that the participants who reported freezing of gait (FoG) would have affected coherence. Since FoG episodes were rarely experienced by the participants during straight-line walking in our experiment, we tried to verify at least whether coherence would be somehow associated with the history of FoG (N-FOGQ) in PD in the OFF state. Additionally, we correlated coherences with MDS-UPDRS-III and TUG. We verified that increases in overall cortico-muscular and intermuscular coherence outcomes during the stance and swing phases were moderately to strongly correlated with improved functional mobility but not correlated with N-FOGQ and MDS-UPDRS-III to the same extent (Fig. 7). Seemingly, the organization of cortical-muscular inputs (inferred by cortico-muscular and intermuscular coherence) may be directly linked to functional mobility (TUG) and, to a lesser extent, to PD's motor manifestation (MDS-UPDRS-III) and history of FoG. Also, a relevant question for future studies would be to verify the link between cortico-cortical, cortico-muscular, and intermuscular coherence with the pathophysiology of FoG since we could only verify the indirect association between coherence and FoG, as we did not have enough events to compute a robust coherence analysis. Another relevant direction for future studies is the development of windows for spectral analysis (e.g., such as coherence) based on EMG signals instead of kinematics or kinetics. This would increase the "ecological" validity of the analysis, as the windows would potentially be more representative in terms of analyzing the window in which a certain muscle is active. However, implementing such methods during gait analysis poses challenges due to the diverse muscular functions, activity patterns, and peaks of muscles, as well as the inherent variability in electrophysiological data across individuals and within steps. Additional relevant directions for future studies also include exploring the clinical meaning of coherence for locomotion in healthy aging and PD, as well as integrative analysis of kinematic and muscle synergies. Such analysis may complement the present study in supporting the understanding of the role of dopamine in the neural control of gait.

In conclusion, our study indicates that Side has, overall, little effect on gait score, on gait harmony, and on cortico-cortical, cortico-muscular, and intermuscular coherence. In spite of the relatively symmetrical gait control, we did observe defective cortico-muscular and intermuscular connectivity during gait in people with PD in the OFF state. The defective cortico-muscular coherences were accompanied by decreased bilateral step-timing coordination, shorter stride length, and slower gait. However, dopamine improved the defective coherence during PD's gait, an effect reflected in better gait performance and better coordination. We interpreted that dopamine corrects defective sensorial feedback and sensory motor integration, resulting in increased cortico-muscular coherence in the alpha bands and improvements in gait.

## Methods
### Participants
In this study, we used an existing database of the Research Center of Advanced Technologies in Rehabilitation (CATR) at Sheba Medical Center, Ramat Gan, Israel. Initially, data sets from 28 people with PD and 10 healthy OA had the potential to be included. Inclusion criteria were diagnosis of idiopathic PD[43], 50–90 years of age, being under levodopa treatment, able to walk unassisted and without pain for at least 100 m, and participation in the protocol during ON and OFF conditions (in two separate visits, 2–3 weeks apart). During data collection, the exclusion criteria were as follows: surgery within the last 6 months or brain surgery at any point in the past; history of stroke: severe peripheral neuropathy, with symptomatic lumbar spinal stenosis; and serious co-morbidities that affect gait and capacity to perform the protocol. Since the present study relies on electroencephalography (EEG) and surface electromyography (sEMG) data, additional exclusion criteria related to the data analysis were adopted, such as rejection of EEG channels C3 or C4 during signal preprocessing, higher cortico-cortical coherence in all bands (indicative of cross-talk), and lastly, high coherence that was not consistent with near-zero lag synchronization (suggesting cross-talk)[44]. After applying EEG and sEMG data-related exclusion criteria, the final sample was comprised of 14 people with PD and 9 OA

(Supplementary Table 1). The experimental protocol was approved by the Sheba Medical Center Institutional Review Board. All participants provided written informed consent prior to entering the study.

## Experimental procedure

The experimental procedures involved two visits in two medication states (OFF and ON), separate visits 2–3 weeks apart. OFF visits occurred first, and the participants were considered in the OFF state if they had been without anti-parkinsonian intake for approximately 12 h. ON visits were planned with reference to the daily medication intake of the participant and were thus allowed for the examination of the people with PD during a period that included the medication's peak effect. OA were invited only to one visit.

Demographic characteristics, global cognition, and motor PD impairments (at the start of each visit—ON and OFF) were evaluated by using questionnaires, the Montreal Cognitive Assessment (MoCA, Hebrew validated version[45]), and the motor part (III) of MDS-UPDRS[46], respectively. For each side, the sum of specific items of the MDS-UPDRS (items 3.3, 3.4, 3.5, 3.6, 3.7, 3.8, 3.15, 3.16, and 3.17) was used to determine the more (higher score) and less (lower score) affected side (MAS and LAS, respectively). In addition, functional mobility was assessed via TUG[47], and, for PD, the LEDD was computed[48]. Whether participants suffered from FoG during their daily routines was also assessed; they were deemed to suffer from FoG if they scored a 3 on question 2 in Part II of the New Freezing of Gait Questionnaire (N-FOGQ)[49], and/or if the neurologist of the participant reported a history of FoG. However, we only considered gait events in which FoG was absent (detailed below).

Regarding gait assessment, the participants performed gait trials that included straight-line walking, i.e., continuous back and forth, between two cones placed 12 m apart (Fig. 9a) in a 3-m-wide corridor with single trials lasting 3–4 min. Between trials, participants were allowed rest pauses as needed, and the gait assessment lasted 15–30 min based on the fatigue level of the participants. The participants were asked to walk at their self-selected comfortable pace. Prior to the trials, we made technical preparations in order to register cortical activity, muscle activity, and spatial and temporal stride outcomes (e.g., stride length, duration, and gait speed), and gait events (Fig. 9a).

We collected cortical activity using a 32-channel portable EEG cap (EEGO Sports™, eemagine Medical Imaging Solutions GmbH, Berlin, Germany). During the experiments, however, we collected EEG data from either 32 or 19 EEG electrodes (the latter cases were due to technical issues). Cortical electrical activity was recorded at a sampling rate of 2048 Hz. EEG and sEMG were acquired using the same device and, thus, automatically synchronized. For time-synchronization between EEG and sEMG with the OPAL system (details below), we performed simultaneous tapping in both systems, generating signals in both systems that could be identified post hoc. We also used video recordings of those tapping events and gait. Post-hoc video analyses were carefully performed to annotate synchronization/delays and further exclude freezing episodes, intentional stops, and 180° turns (turns around the cones that determined the back and forth of walking—Fig. 9). Video analyses (annotation files) were also used to confirm systematic synchronization in the systems.

We registered sEMG using a 4-channel per leg-muscle sEMG recording system (Eego referential amplifier, eemagine Medical Imaging Solutions GmbH, Berlin, Germany), which was the same EEG system used. sEMG recordings were taken at a frequency rate of 2048 Hz from the vastus lateralis (vastus), biceps femoris (biceps), gastrocnemius lateralis (gastrocnemius), and tibialis anterior (tibialis). To control for cross-talk effects, we also recorded sEMG signals using specific contractions according to the muscle involved. sEMG electrodes (bipolar hydrogel surface electrodes, 10 mm in diameter, 24 mm apart) were placed parallel to the muscle fibers according to the guidelines provided in "Surface Electromyography for the Non-Invasive Assessment of Muscles (SENIAM)"[50] and the International Society of electrophysiology and Kinesiology (ISEK)[51] recommendations.

To record the movement kinematics and identify gait events (toe-offs and heel strikes), the participants wore 6 OPAL sensors (OPAL, APDM

Wearable Technologies INC., Portland, Oregon, USA). The OPAL sensors consist of wireless, synchronized, triaxial accelerometers and gyroscopes that record body motion at 128 Hz. The sensors were placed, with Velcro straps, bilaterally on the subjects' shanks and wrists, sternum, and lumbar (L5). This gait analysis system provided the gait cycle events (heel strike and toe-off timing) that were used in this study (Mobility Lab software, www.apdm.com)[52,53].

## Outcome measures and data analyses

Data analysis was performed using MATLAB (v.22a, The MathWorks Inc., Natick, MA). Gait events and stride outcomes, detected via the Mobility Lab software, were acquired from the kinematic data recorded by the OPAL system[52,53]. Only straight-line segments were included in the analysis, i.e., turning segments were automatically identified by gyroscopes of the OPAL sensors and confirmed using video recordings for exclusion. When a FoG event was detected during linear walking via the video recordings, this segment was also excluded from the analysis.

In terms of gait score, spatial-temporal stride outcomes were computed/extracted from the default OPAL data for left and right side steps/MAS and LAS, as follows: stride length (forward distance traveled by a foot during a gait cycle, normalized (%) by the participant's height); swing phase (time (s) of the gait cycle during which the foot is not on the ground); stance phase (time (s) during which the foot is in contact with the ground); stride duration (duration (s) of a full gait cycle—from one foot's initial contact to the consecutive foot contact); speed (forward distance traveled during the gait cycle divided by the gait cycle duration, normalized (%) by the participant's height).

Using gait events, the PCI ([%]—full description in Plotnik et al.[2] and in Supplementary Note 2 and Supplementary Figs. 1 and 2 in the Supplementary Information) was computed as a measure of bilateral coordination of gait. Briefly, PCI is a metric that measures the accuracy and consistency of left–right stepping-phase values (relative to the "ideal" phase of 180°). Higher PCI values indicate worse coordination.

In order to measure whether PD affects the canonical proportion between stance/swing phases that hypothetically should be close to 1.618, as an indication of gait harmony, we computed the Golden Ratio ($\phi$). The $\phi$ was computed using the following formula, which calculates the ratio between the swing and stance stride cycle phases[1]:

$$\phi = \frac{\text{Stance time}}{\text{Swing time}} = \frac{\text{Stride time}}{\text{Stance time}}$$

The ratio between the different gait phases, such as between the stance and swing and between stride and stance phases, in healthy subjects reflects the irrational number $\phi$ (i.e., 1.618, the golden ratio found to be indicative of harmony of gait). The ratio was calculated separately for the left and right/MAS and LAS. Detailed information on the relationship between $\phi$ and gait harmony is described in the Supplementary Note 1 in the Supplementary Information.

EEG data were first preprocessed in MATLAB using the EEGLAB toolbox[54]. Details of EEG preprocessing are described in Koren et al.[10]. Briefly, the following steps were taken: (1) data re-sampling to 256 Hz; (2) signal de-trending by finite impulse response (FIR) high-pass filtering (1.2 Hz cut-off frequency); (3) line noise (and harmonics) removal at 50 Hz (CleanLine method); (4) removal of noisy channels and movement artifacts (MA, ASR method); (5) employing "runica" for the Independent Component Analysis (ICA) in which we removed eye-, neck-, and face-muscle artifact components by visual inspection (components removed = $6.37 \pm 1.31$)[38–42]. For our purpose, we considered the activity registered only on C3 and C4 to presumably represent the left and right motor cortices (M1)[55].

sEMG data were initially down-sampled to 256 Hz to fit with EEG and gait events (heel strike and toe-off) detected via OPAL. sEMG data were visually inspected to minimize noise and artifacts and ensure synchronization. Then, the data were 5-Hz high-pass filtered, using a second-order

Butterworth filter, and full-wave rectified. Despite some deliberation as to whether to rectify sEMG signals in coherence analysis, sEMG signal rectification may better represent the effects of spike timing and more closely match sEMG and MU spectrum information[56,57]. The windows of analysis were determined according to the gait events (heel strike and toe-off), marking the swing and stance phases (Fig. 9b, c). Coherence was computed for each phase. Because of the variability in the proportion of the swing and stance phases to the total gait cycle across and within participants (especially people with PD), sEMG and EEG signals were re-scaled to have 200 data points in each window of analysis (swing and stance). This procedure ensures that coherence is computed within a homogeneous time frame and frequency resolution and that groups' variance within each swing and stance duration would not influence coherence. We excluded values around toe-offs and heel strikes (10 data points prior to and after each event—10% of the window) to minimize possible artifact interference of gait events on coherence.

We calculated coherence to describe frequency-domain couplings between two sources for the swing and stance phases (Fig. 9c), separately for the left and right sides (1) cortico-cortical coherence—C3 and C4; (2) cross-cortico-muscular coherence for right—C4 and left muscles; 3) cross-cortico-muscular coherence for left—C3 and right muscles; and (4) intermuscular coupling—sEMG and sEMG, Fig. 9a). We computed coherence ($C$) based on spectral estimates for EEG and sEMG signals constructed using the discrete Fourier transform of nonoverlapping data segments at a given epoch relative to re-scaled swing and stance phases. Spectral estimates for each epoch were then averaged across steps and used to calculate coherence as the squared modulus of the cross-spectrum normalized by the product of the 2 auto-spectra for each Fourier frequency:

$$C(\lambda) = \frac{|f_{xy}(\lambda)|^2}{f_{xx}(\lambda) \cdot f_{yy}(\lambda)}$$

where $f_{xx}$ and $f_{yy}$ are the auto-spectra of each source, and $f_{xy}$ refers to the cross-spectrum of a pair of sources (Fig. 9d). Coherence functions provide normative measures of linear association on a scale from 0 (absent) to 1 (completely correlated) in a frequency range of 0–55 Hz, and separately organized in alpha (5–15 Hz), beta (16–35 Hz), and gamma bands (36–55 Hz). Coherence is used to quantify the strength and frequency correlation of oscillatory couplings. We considered as sources for computing coherence the following signals separated for each side (left and right steps and further grouped in MAS and LAS in people with PD):

- Cortico-cortical coherence: C3 and C4 EEG channels.
- Cortical-muscular coherence: left steps – C4 (cortical after grouping according to MAS and LAS) and each muscle (tibialis anterior, gastrocnemius lateralis, vastus, and biceps femoris) of the lower left limb; right steps – C3 (cortical after grouping according to MAS and LAS) and each muscle (tibialis anterior, gastrocnemius lateralis, vastus, and biceps femoris) of the lower right limb.
- Intermuscular coherence: antagonistic muscle pairs—vastus and biceps, and tibialis and gastrocnemius; synergistic muscle pairs—vastus and tibialis, and biceps and gastrocnemius for left and right muscle pairs.

For each participant, coherence was considered significant when it surpassed the confidence limit determined based on the number of steps (L), calculated using the formula[58]:

$$1 - (\alpha)^{\frac{1}{L-1}}$$

where $\alpha = 0.05$ and $L$ is the number of steps used in the analysis. During data processing, we ascertained that cumulant density plots were near zero-lag synchronization and that the coherence was not higher than 0.5 across a wide range of frequencies, thus minimizing the likelihood that cross-talk affected coherences[59].

## Statistics and reproducibility

Statistical analyses were performed in SPSS for Windows (Version 25, IBM, Armonk, NY, USA) and R-Studio (Version 2022.02., Boston, USA). After computing the outcomes, we reorganized the individual data in people with PD according to LAS and MAS only for statistical comparison. When Shapiro–Wilk tests revealed non-normal distribution, data were log-transformed for further comparisons using ANOVA. First, to determine the effects of medication, we employed one-way ANOVA as a factor for Medication (ON vs. OFF) for PCI, and two-way ANOVA with Side (MAS vs. LAS) and Medication as within factors for strides outcomes, $\phi$, and coherence. Second, separate one-way and two-way ANOVAs were conducted to verify the effects of PD in comparison to the OA group. These analyses compared people with PD in the ON state (PD-ON) with those in the OA group, as well as people with PD in the OFF state (PD-OFF) with the OA group, considering parameters such as PCI, stride outcomes, $\phi$, and coherence. For the factor of Side in control groups, we compared gait phases from the left with those from the right. When interactions and main effects were significant, post hoc comparisons for each factor were made, and the level of significance ($\alpha = 0.05$) was adjusted for multiple comparisons using Bonferroni correction. For post hoc and $t$ test comparisons, Cohen's $d$ was calculated, and we interpreted 0.21–0.50, 0.51 to 0.79, and >0.79 as small, medium, and large effect sizes ($d$), respectively[60].

Specifically, to verify whether potential Medication- or Side-induced differences in coherence correlate with gait outcomes and gait harmony, we had planned to compute correlations only for gait and coherence outcomes in which ANOVA indicated Side and/or Medication main effects. We did not observe any significant Side-related differences in stride outcomes and harmony (see details in the "Results" section), and thus, we did not compute a correlation between Side differences (see details in the Supplementary Note 3 in the Supplementary Information). ANOVA, however, indicated Medication effects for PCI, step length, speed, and coherences. Thus, we calculated Spearman's correlation for Medication-induced changes on those outcomes (i.e., PCI, step length, speed, and coherences) by averaging the data from LAS and MAS and then computing the delta ($\Delta = ON\ minus\ OFF$) for those outcomes. A secondary correlation was calculated to verify the functional/clinical meaning of absolute neurophysiological coherence. Thus, we computed the correlation between PD clinical characteristics (MDS-UPDRS-III, TUG, LEDD, N-FOGQ) and value coherence only for the PD-OFF state. Because of the relatively small sample size, in addition to the $p$ value ($p > 0.05$), we also interpreted the "strength" of the correlation, and we assumed rho-values = 0–0.19, 0.2–0.39, 0.40–0.59, 0.6–0.79, and 0.8–1 to indicate very weak, weak, moderate, strong, and very strong correlations, respectively (and we only reported results above or equal to moderate).

## Reporting summary

Further information on research design is available in the Nature Portfolio Reporting Summary linked to this article.

## Data availability

The data generated or analyzed in this study are included in this published article and its Supplementary Information (in the form of Supplementary Note, Supplementary Table, and Supplementary Figs.) and Supplementary Data files. The source data used to generate plots and statistical analysis can be found in the Supplementary Data file.

## Code availability

For data analyses, MATLAB scripts (for gait outcomes, for spectral power computation, and coherence) are used in a particular order, as detailed in the "Methods" section of the paper.

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

## Acknowledgements

P.C.Rd.S. was supported by the IDOR/Pioneer Science Initiative (www.pioneerscience.org) and by the postdoctoral fellowship program of Feinberg Graduate School - Weizmann Institute of Science. T.F. was supported by grants from the Estate of Naomi K. Shapiro, the Rolf and Alice Wiklund Parkinson's Disease Research Fund, the Steven Gordon and Levine Foundations, and the Rudolph and Hilda U. Forchheimer Foundation. This study was supported in part by a grant from the Israel Science Foundation (ISF, # 1657–16) and in part by a grant from the Israeli Ministry of Health (MOH, # 3000-14527). The funders played no role in the study design, data collection, analysis and interpretation of data, or the writing of this manuscript.

## Author contributions

P.C.R.d.S., T.F. and M.P. conceptualized and designed the study. B.H. was responsible for the data acquisition. P.C.R.d.S., O.K. and B.H. worked on data analysis. P.C.R.d.S. and M.P. interpreted the results. P.C.R.d.S. drafted the first version of the manuscript. T.F. and M.P. revised the article critically for important intellectual content. P.C.R.d.S., T.F., B.H., O.K. and M.P. read and approved the final version and agreed to be accountable for all aspects related to the accuracy or integrity of the work.

## Competing interests

The authors declare no competing interests.
