## [Peer Review File · Communications Biology]

Reviewers' comments:

Reviewer #1 (Remarks to the Author):

This is a revised version of a manuscript that I previously reviewed. The authors have addressed several issues raised by the previous reviewers. However, the revision seems somewhat superficial. Specifically, in the opinion of this reviewer, the authors have not sufficiently focused the paper or improved its readability. This primarily concerns the presentation of data, but also encompasses language-related issues. While the data may hold value for a specialist audience, the overall value of the present manuscript seems limited for a broader readership.

Regarding specific concerns raised previously, the authors have misconstrued my argument. Coherence analyses are categorized into specific phases of gait (stance, swing), each of which is defined kinematically. Since corticomuscular coherence pertains to muscular activity, it might be more enlightening to utilize muscular activity to distinguish between different phases. This contention is distinct from the matter of how the analysis of muscle synergies might enhance our comprehension of freezing of gait, as now introduced by the authors.

As episodes of FOG were excluded from the analysis, any finding of corticomuscular coherence deviations can only very indirectly be related to the pathophysiology of FOG.

Reviewer #2 (Remarks to the Author):

This is reviewer #4 from previous review. The authors have responded to all my comments, the restructured introduction with clear hypotheses and the revised discussion with summaries at the start of each section give an improved narrative for the manuscript.

I would change the phrase "In cortico-cortical brain areas,..." in Introduction , page 2, line 79, perhaps "In cortical areas, ..."

We thank the reviewers for their time and insightful comments on improving the manuscript. We have addressed/answered each criticism and indicated the changes we made in the manuscript (blue).

Reviewers' comments:

Reviewer #1 (Remarks to the Author):

This is a revised version of a manuscript that I previously reviewed. The authors have addressed several issues raised by the previous reviewers. However, the revision seems somewhat superficial. Specifically, in the opinion of this reviewer, the authors have not sufficiently focused the paper or improved its readability. This primarily concerns the presentation of data, but also encompasses language-related issues. While the data may hold value for a specialist audience, the overall value of the present manuscript seems limited for a broader readership.

RESPONSE: Thank you for your comments, which certainly helped improve the quality of the manuscript. We sent the manuscript for language editorial service, and we believe that this process helped improve the presentation of the data and language-related issues. In the tracked version, we highlighted in blue some of the changes provided by the service. We also selected to report cortico-cortical, cortico-muscular, and intermuscular coherence in the alpha, beta, or gamma band instead of cortico-cortical, cortico-muscular, and intermuscular alpha, beta, or gamma coherence. We edited the figures by increasing their font sizes and their qualities to improve the presentation of the data. We also changed segments that the language service identified as unclear or complex to follow (highlighted in blue in the manuscript track version).

Regarding specific concerns raised previously, the authors have misconstrued my argument. Coherence analyses are categorized into specific phases of gait (stance, swing), each of which is defined kinematically. Since corticomuscular coherence pertains to muscular activity, it might be more enlightening to utilize muscular activity to distinguish between different phases. This contention is distinct from the matter of how the analysis of muscle synergies might enhance our comprehension of freezing of gait, as now introduced by the authors.

RESPONSE: First, we would like to acknowledge that we did not respond properly to this comment during the previous revision round. We focused on potentially addressing 'muscle synergies' rather than on the method of timing the gait phases (i.e., EMG vs 'kinematics').

We thank you for this interesting suggestion for window coherence analysis to EMG burst instead of kinematics. However, there are several challenges and limitations in determining gait phases based on EMG, as follows:

- 1) Electrophysiological data, in general, and EMG, in particular, have a considerably variable pattern during cyclic movement within and between individuals and muscles. Thus, participants from the different groups (PD and Older) and medication conditions (ON vs. OFF) may have even greater variability in patterns (i.e., when in the cycle the muscles are active), peaks (amplitude), and in the temporal features of the EMG burst during walking that is not time-locked. Thus, EMG-based windowing (segmentation) of the data would lead to misleading results, conclusions. The reason is that the basis of computation of coherence would differ between participants per se AND in relation to a determined cycle. Potentially, we would include data from one participant/cycle that would cancel

data from another participant/cycle. Therefore, we particularly do not know whether to account for this variability to create a robust method that allows us to predict and identify events during gait based on EMG signals, and we are not familiar with any study that proposed to compute coherence during gait based on the EMG window (listed below).

2) Differently, kinematics, specifically considering OPAL [(Mancini et al. 2011; Spain et al. 2012; Morris et al. 2019) see other at <https://apdm.com/publications/>], and kinetics (e.g., force plate data) is a more robust and validated method to determine gait events and metrics. With such methods, we can also rely on specific events to occur independent of the individual variability in muscle activation without a greater variation in time. Therefore agreeing with an extensive number of studies ((Halliday et al. 2003; Norton and Gorassini 2006; Nielsen et al. 2008; Petersen et al. 2012; Artoni et al. 2017; Roeder et al. 2018, 2020; Spedden et al. 2019; Jensen et al. 2019; Yokoyama et al. 2020; Gennaro and Bruin 2020; Santos et al. 2020, 2022; Weersink et al. 2021b, a; Sato and Choi 2022; Feng et al. 2023), we argue that computing coherence analysis by windowing the data relative to kinematically and kinetically determined gait cycle instead of EMG burst onset/offset represents a more functional way and facilitate the interpretation and comparison with the literature.

3) Finally, although reconstructed data based on principal components from health subjects theoretically suggest a relative pattern in muscle activation (e.g., gastrocnemius activate in specific periods of gait cycle - near to the toe-off), the selection would be arbitrary considering that the muscles analyzed are predominantly active at different times during the gait cycle (e.g., Tibial is active around the heel strike, and Rectus femoris activates mainly after the heel strike). Therefore, considering classifying the cycle based on one muscle does not necessarily ensure the activation of another muscle at the same window. This would not only interfere in determining cycles and the functional interpretation (or clinical meaning) but also in selecting the window for computing intermuscular coherence.

We however recognize that it might be relevant to future studies to develop robust methodologies in a way that allows us to predict and determine windows of analysis based on EMG burst (onset and offset) for computing spectral analysis such as coherence. We included the following statement in the limitation of the study:

“Another relevant direction for future studies is the development of windows for spectral analysis (e.g., such as coherence) based on EMG signals instead of kinematics or kinetics. This would increase the “ecological” validity of the analysis, as the windows would potentially be more representative in terms of analyzing the window in which a certain muscle is active. However, implementing such methods during gait analysis poses challenges due to the diverse muscular functions, activity patterns, and peaks of muscles, as well as the inherent variability in electrophysiological data across individuals and within steps.”

As episodes of FOG were excluded from the analysis, any finding of corticomuscular coherence deviations can only very indirectly be related to the pathophysiology of FOG.

RESPONSE: Thank you for your suggestion. We completely agree with the reviewer.

We did have enough FOG events for performing a robust coherence analysis (the resolution/validity of the analysis increases according to the increase in the number of segments (steps/events) analyzed (Halliday et al. 1995)), and thus, we decided to exclude the events. Additionally, we agree that we can

only indirectly relate the analysis with the pathophysiology of FOG. We reviewed the entire manuscript to ensure that we can only indirectly relate this to FOG, and we included a new segment in the future direction paragraph.

“Also, a relevant question for future studies would be to verify the link between cortico-cortical, cortico-muscular, and intermuscular coherence with the pathophysiology of FoG since we could only verify the indirect association between coherence and FoG, as we did not have enough events to compute a robust coherence analysis.”

REFERENCES

- Artomi F, Fanciullacci C, Bertolucci F, et al (2017) Unidirectional brain to muscle connectivity reveals motor cortex control of leg muscles during stereotyped walking. *Neuroimage* 159:403–416. <https://doi.org/10.1016/J.NEUROIMAGE.2017.07.013>
- Feng HS, Jiang YN, Lin JP, et al (2023) Cortical activation and functional connectivity during locomotion tasks in Parkinson’s disease with freezing of gait. *Front Aging Neurosci* 15:1068943. <https://doi.org/10.3389/FNAGI.2023.1068943/BIBTEX>
- Gennaro F, Bruin ED (2020) A pilot study assessing reliability and age-related differences in corticomuscular and intramuscular coherence in ankle dorsiflexors during walking. *Physiol Rep* 8:. <https://doi.org/10.14814/phy2.14378>
- Halliday DM, Conway BA, Christensen LOD, et al (2003) Functional Coupling of Motor Units Is Modulated During Walking in Human Subjects. *J Neurophysiol* 89:960–968. <https://doi.org/10.1152/jn.00844.2002>
- Halliday DM, Rosenberg JR, Amjad AM, et al (1995) A framework for the analysis of mixed time series/point process data-Theory and application to the study of physiological tremor, single motor unit discharges and electromyograms. *Prog. Biophys. Mol. Biol.*
- Jensen P, Frisk R, Spedden ME, et al (2019) Using Corticomuscular and Intermuscular Coherence to Assess Cortical Contribution to Ankle Plantar Flexor Activity During Gait. *J Mot Behav* 51:668–680. <https://doi.org/10.1080/00222895.2018.1563762>
- Mancini M, King L, Salarian A, et al (2011) Mobility Lab to Assess Balance and Gait with Synchronized Body-worn Sensors. *J Bioeng Biomed Sci Suppl* 1: <https://doi.org/10.4172/2155-9538.S1-007>
- Morris R, Stuart S, Mcbarron G, et al (2019) Validity of MobilityLab (version 2) for gait assessment in young adults, older adults and Parkinson’s disease. *Physiol Meas* 40:095003. <https://doi.org/10.1088/1361-6579/AB4023>
- Nielsen JB, Brittain JS, Halliday DM, et al (2008) Reduction of common motoneuronal drive on the affected side during walking in hemiplegic stroke patients. *Clin Neurophysiol.* <https://doi.org/10.1016/j.clinph.2008.07.283>
- Norton JA, Gorassini MA (2006) Changes in Cortically Related Intermuscular Coherence Accompanying Improvements in Locomotor Skills in Incomplete Spinal Cord Injury. *J Neurophysiol* 95:2580–2589. <https://doi.org/10.1152/jn.01289.2005>
- Petersen TH, Willerslev-Olsen M, Conway BA, Nielsen JB (2012) The motor cortex drives the muscles during walking in human subjects. *J Physiol* 590:2443–2452. <https://doi.org/10.1113/jphysiol.2012.227397>
- Roeder L, Boonstra TW, Kerr GK (2020) Corticomuscular control of walking in older people and people with Parkinson’s disease. *Sci Rep* 10:2980. <https://doi.org/10.1038/s41598-020-59810-w>
- Roeder L, Boonstra TW, Smith SS, Kerr GK (2018) Dynamics of corticospinal motor control during overground and treadmill walking in humans. *J Neurophysiol* 120:1017–1031. <https://doi.org/10.1152/jn.00613.2017>
- Santos PCR, Lamothe CJC, Barbieri FA, et al (2020) Age-specific modulation of intermuscular beta coherence during gait before and after experimentally induced fatigue. *Sci Rep* 10:. <https://doi.org/10.1038/s41598-020-72839-1>
- Santos PCR, Zijdwind I, Lamothe C, et al (2022) Walking speed does not affect age-differences in

- ankle muscle beta-band intermuscular coherence during treadmill walking. *Brazilian J Mot Behav* 16:155–163
- Sato SD, Choi JT (2022) Corticospinal drive is associated with temporal walking adaptation in both healthy young and older adults. *Front Aging Neurosci* 0:900. <https://doi.org/10.3389/FNAGI.2022.920475>
- Spain RI, St. George RJ, Salarian A, et al (2012) Body-worn motion sensors detect balance and gait deficits in people with multiple sclerosis who have normal walking speed. *Gait Posture* 35:573–578. <https://doi.org/10.1016/J.GAITPOST.2011.11.026>
- Spedden ME, Choi JT, Nielsen JB, Geertsen SS (2019) Corticospinal control of normal and visually guided gait in healthy older and younger adults. *Neurobiol Aging* 78:29–41. <https://doi.org/10.1016/j.neurobiolaging.2019.02.005>
- Weersink JB, de Jong BM, Halliday DM, Maurits NM (2021a) Intermuscular coherence analysis in older adults reveals that gait-related arm swing drives lower limb muscles via subcortical and cortical pathways. *J Physiol* 599:2283–2298. <https://doi.org/10.1113/JP281094>
- Weersink JB, De Jong BM, Maurits NM (2021b) Neural coupling between upper and lower limb muscles in Parkinsonian Gait. *Clin Neurophysiol*. <https://doi.org/10.1016/j.clinph.2021.11.072>
- Yokoyama H, Yoshida T, Zabjek K, et al (2020) Defective corticomuscular connectivity during walking in patients with Parkinson’s disease. *J Neurophysiol* 124:1399–1414. <https://doi.org/10.1152/jn.00109.2020>

Reviewer #2 (Remarks to the Author):

This is reviewer #4 from previous review. The authors have responded to all my comments, the restructured introduction with clear hypotheses and the revised discussion with summaries at the start of each section give an improved narrative for the manuscript.

RESPONSE: We thank Reviewer #2 for their contribution to improving the quality of the manuscript.

I would change the phrase “In cortico-cortical brain areas,...” in Introduction , page 2, line 79, perhaps “In cortical areas, ...”

RESPONSE: We addressed this suggestion.

REVIEWERS' COMMENTS:

Reviewer #1 (Remarks to the Author):

The authors have addressed all issues raised by this reviewer.